# On the Dual Problem of Convexified Convolutional Neural Networks

**Site Bai**                                                          *bai123@purdue.edu*
*Department of Computer Science*
*Purdue University*

**Chuyang Ke**                                                          *cke@purdue.edu*
*Department of Computer Science*
*Purdue University*

**Jean Honorio**                                                *jhonorio@unimelb.edu.au*
*School of Computing and Information Systems*
*The University of Melbourne*

**Reviewed on OpenReview:** *https://openreview.net/forum?id=OyMuNezwJ1*

## Abstract

We study the dual problem of convexified convolutional neural networks (DCCNNs). First, we introduce a primal learning problem motivated by convexified convolutional neural networks (CCNNs), and then construct the dual convex training program through careful analysis of the Karush-Kuhn-Tucker (KKT) conditions and Fenchel conjugates. Our approach reduces the computational overhead of constructing a large kernel matrix and more importantly, eliminates the ambiguity of factorizing the matrix. Due to the low-rank structure in CCNNs and the related subdifferential of nuclear norms, there is no closed-form expression to recover the primal solution from the dual solution. To overcome this, we propose a highly novel weight recovery algorithm, which takes the dual solution and the kernel information as the input, and recovers the linear weight and the output of convolutional layer, instead of weight parameter. Furthermore, our recovery algorithm exploits the low-rank structure and imposes a small number of filters indirectly, which reduces the parameter size. As a result, DCCNNs inherit all the statistical benefits of CCNNs, while enjoying a more formal and efficient workflow.

## 1 Introduction

In the past decade, convolutional neural networks (CNNs) have become the cornerstone of various deep learning architectures. The performance of CNNs is so impressive, that they have become the standard methods on many tasks such as biomedical imaging (Suzuki, 2017; Yamashita et al., 2018), robot perception (Han et al., 2016; Zhang et al., 2019) and signal processing (Hershey et al., 2017; Eren et al., 2019). Despite the great success in a broad range of applications, the training of CNNs is in general an NP-hard problem (Blum & Rivest, 1992). While in practice gradient-based optimization methods are used in the training process most of the time, there is no theoretical guarantee of achieving global optimality or bounding the rate of convergence.

To tackle the limitations of nonconvex landscapes, Zhang et al. (2017) proposes a class of convex relaxations of CNNs, namely the *convexified convolutional neural networks* (CCNNs). Employing a convex optimization formulation, CCNNs can be optimized efficiently and their statistical properties can be analyzed rigorously (Zhang et al., 2017). In CCNNs, an activation function is induced by a kernel function $\mathcal{K}$. The kernel

function produces a kernel matrix $K$, which needs to be approximately factorized into a matrix $Q$ such that $K \approx QQ^\top$. After that, the CCNN algorithm solves a nuclear norm constrained convex optimization problem, and computes a low rank approximation, which outputs the weights of the network. While stochastic gradient descent (SGD) is still the state-of-the-art, the whole convexified approach allows for efficient and optimal training of the network, and from a theoretical point of view makes precise statistical analysis viable (Zhang et al., 2017).

However on the algorithmic front, the CCNN framework, while enjoying the aforementioned benefits, does come with drawbacks and its own limitations.

- The approximate factorization of the kernel matrix $K \approx QQ^\top$ is not unique. One can use a tall matrix $Q$ either from the Cholesky decomposition of $K$, from $K^{1/2}$, or from $UD^{1/2}$ where $K = UDU^\top$, to name a few. In the latter case, one can also use $UD^{1/2}V^\top$ for any orthonormal $V$, meaning that there exists infinitely many possible $Q$'s! There is no guarantee that every choice of $Q$ performs equally well in the learning task, and thus the factorization problem itself can be tricky. Furthermore, the idea of factorizing the kernel matrix is not suitable from a statistical perspective, given that for CCNNs, kernels with infinitely dimensional basis functions are used (for example, Gaussian RBF kernel).

- Solving the exact kernel factorization problem itself can be computationally expensive. Looking at Algorithm 1 in the CCNN framework (Zhang et al., 2017), the dimension of the kernel matrix is equal to the number of samples $n$ times the number of convolution operations (patches) $p$. From a space complexity perspective, the factorization step alone occupies memory in the order of $O(n^2 p^2)$. This is hardly feasible: for a small network training on 1000 $28 \times 28$ images with stride 1, i.e. 784 convolution operations, with 64 bit double precision, the kernel matrix takes more than 4 terabytes of memory! Similarly, the computational complexity of many factorization algorithms (for example, Cholesky) can take $O(n^3 p^3)$, which can be prohibitive in practice.

- The CCNN algorithm heuristically cut the first $r$ columns of the convolutional weight recovered for the final low rank approximation step, with the number of filters $r$ as a hyperparameter. Setting the hyperparameter too large increases the parameter size with unnecessary information, and setting it too small voids the optimality guarantee. And such impact may accumulate as the number of layers increases.

In this paper, we study the dual framework for learning CCNNs, which we call *dual convexified convolutional neural network* (DCCNN). We first carefully construct the DCCNN learning program by analyzing nuclear norms and Fenchel conjugates. We then solve the convex training problem in the derived dual form. Unfortunately, due to the nature of the sub-differential of the nuclear norm, there is no closed-form expression for us to recover the primal solution from the dual solution. To overcome this, we design a novel weight recovering algorithm that leverages the KKT optimality conditions and the sub-differential of nuclear norm. Without getting the primal solution, our algorithm directly recovers the linear weight and the convolution output using the dual solution and a kernel generating matrix, which applies to kernels with infinitely dimensional basis.

Our DCCNN framework enjoys many benefits on top of the CCNN approach. First, compared to the CCNN approach, our method does not require any approximate factorization of the kernel matrix in the form of $K \approx QQ^\top$. Instead, by solving the optimization problem in DCCNN, we can directly recover the proper solution without introducing any ambiguity or compromise. Next, our approach does not construct the full matrix $K$ for all samples in memory. Our weight recovering algorithm computes entries of the kernel matrix on the fly, greatly reducing the memory overhead. More than that, our DCCNN framework does not require specifying the low rank hyperparameter manually. Our weight recovery algorithm automatically computes the minimum necessary rank from the dual solution, which better exploits the benefits of the low-rankness introduced by the nuclear norm constraint, as it encourages a small number of filters indirectly and reduces the parameter size. To sum up, our DCCNN method inherits all the benefits of convexified CNNs like generalization error and sample complexity bounds, while enjoying a much more formal and efficient workflow without introducing ambiguity.

**Related works.** The area of convex convolutional neural networks is fairly new. In Zhang et al. (2017), it is shown that a class of CNNs with reproducing kernel Hilbert space (RKHS) filters contains a low-rank matrix structure, which can be further convexified by using a nuclear norm constraint and reduced to the

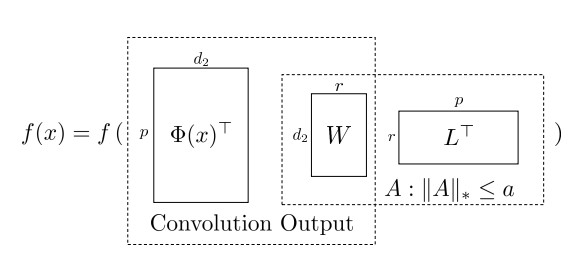
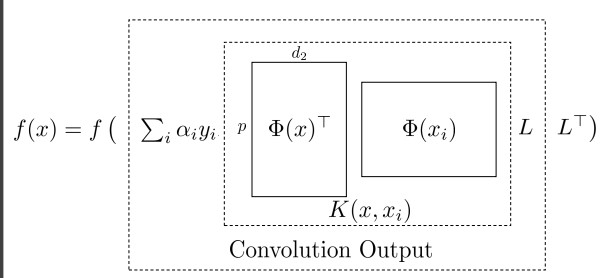

(a) The Primal Framework            (b) The Dual Framework

Figure 1: (a): In the primal framework, the basis function matrix $\Phi(x) = Q$ from the approximate factorization of kernel matrix such that $K \approx QQ^\top$. The convolutional weight $W$ and linear weight $L$ are multiplied together as matrix $A$ with low-rankness enforced by nuclear norm constraint. $W$ is recovered by a low-rank approximation from optimized $A$. (b): The dual framework uses $K(x, x_i)$ without ambiguous factorization, and recovers the weights with the optimized dual variable $\alpha$. The primal solution $A$ cannot be directly recovered because $A$ has no closed-form expression of $\alpha$. Therefore, the dual framework recovers linear weight $L$ and computes the convolution output $\Phi(x)^\top W$ directly without $W$ or $\Phi(x)$.

class of CCNNs. This provides a basis for our work. Other convex neural network researches are more tangential. Mairal et al. (2014) proposed to approximate the CNNs with a translation-invariant kernel. Zhong et al. (2017) analyzed the convergence properties of CNNs. Gunasekar et al. (2018) analyzes a linear version of convolutional networks. Amos et al. (2017) and Makkuva et al. (2020) study the so-called input convex neural networks, where the output is a convex function of the inputs. Arora et al. (2019) studies the exact computation of CNNs with infinite many filters. Ergen & Pilanci (2020) proposes equivalent convex regularizers to the CNN architectures. See Appendix H for further discussion.

**Summary of our contribution.** We provide the following series of novel results in this paper:

- We provide a dual convexified convolutional neural network (DCCNN) framework for learning a class of CNNs. We rigorously construct a dual convex training program through careful analysis of the KKT conditions and Fenchel conjugates. With the dual solution, we also propose a linear weight and implicit convolutional weight recovery algorithm.

- Our DCCNN framework is theoretically formal. Compared to prior literature, our method does not require any ambiguous factorization of the kernel matrix. Through the novel weight recovery algorithms, we directly recover the proper solution. In addition, our approach implicitly encourages a small number of filters, reducing the number of weight parameters without enforcing the low-rankness by heuristics.

- Our DCCNN framework is efficient. It operates without loading the full kernel matrix for all samples in memory, or introducing any unnecessary memory overhead of factorization. As a result, DCCNN inherits all the statistical benefits of convexified CNNs (Theorem 1 in Zhang et al. (2017)), while enjoying a more efficient workflow (See Appendix G.4).

- Our analysis in the DCCNN framework is novel. To derive the dual training program, the nuclear norm in the primal problem requires a careful construction of the Lagrangian through Fenchel conjugates. Furthermore, there is no closed-form expression that can be used to recover the primal solution from the dual solution directly because of the sub-differential of the nuclear norm. Without getting the primal solution, our algorithm directly recovers the linear weight and the convolution output using the dual solution and a kernel generating matrix, which applies to kernels with infinite-dimensional basis.

## 2 Preliminaries

### 2.1 Notation

$\forall n \in \mathbb{N}$, we use $[n]$ to represent the set $\{1, 2, ..., n\}$. We use regular lower-case letters (e.g., $a$, $c$) to denote constant. We use bold lower-case letters like $\mathbf{x}$ to represent vectors. We use capital letters (e.g., $U$, $V$) to represent matrix. We use $\rho$ as the notation for activation function, $\sigma$ as matrix singular values, and $\lambda$ as matrix eigenvalues. $\|\cdot\|_*$ denotes the matrix nuclear norm, and $\|\cdot\|_2$ represents the spectral norm for a matrix, and the Euclidean norm of a vector. We use $\mathrm{Tr}(\cdot)$ to represent the trace of a matrix. For matrix $V$, we use $V(i)$ to represent the $i^{th}$ column vector and $V_{ij}$ to represent the entry in the $i^{th}$ row and $j^{th}$ column. For column vectors $\mathbf{x}_i \in \mathbb{R}^d$ with $i \in [n]$, we denote $[\mathbf{x}_i] \in \mathbb{R}^{d \times n}$ as the concatenation of these column vectors into a matrix. For matrix $U$ and $V$ with the same number of rows, $[U \mid V]$ represents the concatenation of the matrices. We use $\mathbf{I}$ to represent the identity matrix and $\mathbf{0}$ the zero matrix.

### 2.2 Formulation of Convolutional Neural Networks

Given a dataset $\{(\mathbf{x}_1, y_1), ..., (\mathbf{x}_n, y_n)\}$, $\mathbf{x}_i \in \mathbb{R}^{d_0}$, $y_i$, $i \in [n]$, a two-layer CNN, also called one-hidden-layer CNN, is a function that maps the data to the targets by passing the data through a convolutional layer and a linear layer.

The convolutional layer performs convolution operations, in which a subset of $d_1$ entries in $\mathbf{x}_i$ is multiplied element-wise with one of the $r$ convolutional filters $\mathbf{w}_k \in \mathbb{R}^{d_1}$, $k \in [r]$. We call this subset of data entries a *patch* generated from $\mathbf{x}_i$, denoted as $\mathbf{z} = z(\mathbf{x}_i) \in \mathbb{R}^{d_1}$. The number of patches relies on the choice of filter width, padding type, stride, etc., and we use $p$ to denote the total number of patches generated by a sample. The $p$ patches form the patch matrix $Z = [\mathbf{z}_1, ..., \mathbf{z}_p] \in \mathbb{R}^{d_1 \times p}$, and the $r$ convolutional filters form the weight matrix $W = [\mathbf{w}_1, ..., \mathbf{w}_r] \in \mathbb{R}^{d_1 \times r}$. Then the convolutional operations can be represented by the matrix multiplication between the patch matrix and weight matrix, i.e., $Z^\top W$.

The result of the convolutional layer is then passed into an element-wise activation function $\rho(\cdot)$. After that, $\rho(Z^\top W)$ is passed into a linear layer. We denote the weight of the linear layer as $L \in \mathbb{R}^{p \times r}$. For all $j \in [p]$ and $k \in [r]$, $L_{jk}$ is the coefficient multiplied with the $k^{th}$ convolutional filter and the $j^{th}$ patch. Then the output of the two-layer CNN can be represented by $\mathrm{Tr}\left(\rho(Z^\top W)L^\top\right)$.

### 2.3 Convexified Convolutional Neural Networks

**CCNN Formulation with Linear Activation.** Consider a linear activation function, to begin with, the expression for CNN simplifies to $\mathrm{Tr}\left(Z^\top W L^\top\right)$. Define parameter matrix $A = W L^\top \in \mathbb{R}^{d_2 \times p}$. Then we can write the two-layer CNN using the new parameter $A$. As the multiplication of the convolutional weight and linear weight matrices, $A$ is intrinsically low-rank, i.e., $\mathrm{rank}(A) \leq r$. Zhang et al. (2017) enforces the low-rankness by an extra relaxed constraint to bound the nuclear norm of $A$. Therefore CCNN takes the form $\mathrm{Tr}\left(Z^\top A\right)$, in which $\|A\|_* \leq a$ for some constant $a$.

**Activation and Reproducing Kernel Hilbert Space.** As proposed by Zhang et al. (2017) (Lemma 1 and Lemma 2 in their paper) and later generalized by Bietti & Mairal (2019), CNNs with certain choices of activation function are contained in some reproducing kernel Hilbert space (RKHS). In other words, for some specific activation function $\rho$, there is a corresponding kernel function $\mathcal{K}(\cdot, \cdot) = \phi(\cdot)^\top \phi(\cdot)$ that produces similar non-linearity. The basis function $\phi(\cdot) : \mathbb{R}^{d_1} \to \mathbb{R}^{d_2}$ could be a mapping to an infinite dimensional vector space, i.e., $d_2$ could be infinity.

**CCNN Formulation with Non-linear Activation.** For the basis function $\phi(\cdot)$, define basis function matrix

$$\Phi(\mathbf{x}_i) = \left[\phi(z_1(\mathbf{x}_i)), \cdots, \phi(z_p(\mathbf{x}_i))\right] \in \mathbb{R}^{d_2 \times p},$$

and with convolution weight matrix $W \in \mathbb{R}^{d_2 \times r}$, the convex relaxation of non-linear activated CNN can be represented with the kernel basis function, which leads to a convexified two-layer CNN function:

$$f(\mathbf{x}_i) = \mathrm{Tr}\left(\Phi(\mathbf{x}_i)^\top W L^\top\right). \tag{1}$$

In other words, activating the convolution output $\rho(Z^\top W)$ is equivalent to passing the data through the kernel basis function, i.e., $\Phi(\mathbf{x}_i)^\top W$. Define parameter matrix $A = W L^\top \in \mathbb{R}^{d_2 \times p}$. Then we can write the two-layer CNN using the new parameter $A$:

$$f(\mathbf{x}_i) = \mathrm{Tr}\left(\Phi(\mathbf{x}_i)^\top A\right). \tag{2}$$

**Learning CCNN.** In this paper we consider the task of learning two-layer CNN functions for classification with some loss $\ell(\cdot)$ that is convex and non-increasing, e.g. hinge loss, logistic loss, exponential loss. Using the convexified CNN function defined in Eq. (2), the parameter matrix $A$ can be learned by solving the following optimization problem. For binary classification,

$$\hat{A} = \arg\min_A \|A\|_* + c \sum_{i=1}^n \ell\left(y_i \mathrm{Tr}\left(\Phi(\mathbf{x}_i)^\top A\right)\right) \tag{3}$$

where $c > 0$ is a hyperparameter. Furthermore, in the context of $m$-class classification, there is a predictor for each class, i.e., $f_k(\mathbf{x}_i) = \mathrm{Tr}\left(\Phi(\mathbf{x}_i)^\top A_k\right)$ for every $k \in [m]$. Correspondingly, the parameter $A = [A_1, ..., A_m]$ can be learned by minimizing the loss for multi-class classification:

$$\hat{A} = \arg\min_A \|A\|_* + c \sum_{i=1}^n \sum_{k \neq y_i} \ell\left(f_{y_i}(\mathbf{x}_i) - f_k(\mathbf{x}_i)\right) \tag{4}$$

It is worth noticing that the $\Phi(\mathbf{x}_i)$ appearing in the problem can be infinitely dimensional, and can only be represented by some heuristic finite approximation in practice, which consequently introduces ambiguity. To tackle this problem, we consider the dual formulation in this paper.

**Layerwise Training.** The CCNN formulation corresponds to the architecture of one convolutional layer and one linear layer. To extend the framework to deeper architectures, the layerwise training technique (Zhang et al., 2017; Belilovsky et al., 2019) is applied. The convolutional layers are trained one layer at a time, taking the output of the optimized previous layer as input, concatenated with one linear layer that takes part only in the training of the one convolutional layer.

## 3 Main Results

We begin by introducing the primal optimization problem. Learning a two-layer convexified CNN, i.e., solving the loss minimization in Eq. (3) is equivalent to solving the following optimization problem:

$$\begin{aligned} \underset{A}{\text{minimize}} \quad & \|A\|_* + c \sum_{i=1}^n \ell(\xi_i) \\ \text{subject to} \quad & y_i \mathrm{Tr}\left(\Phi(\mathbf{x}_i)^\top A\right) \geq \xi_i, \ \forall i \in [n], \end{aligned} \tag{5}$$

in which $\xi_i$ is an equivalent variable of $y_i \mathrm{Tr}\left(\Phi(\mathbf{x}_i)^\top A\right)$, and $c > 0$ is some hyperparameter. $\ell(\cdot)$ is some convex and non-increasing loss function. In fact, many common loss functions fall into this category, e.g. hinge loss $\ell_H(x) = \max(0, 1 - x)$, squared hinge loss $\ell_{SH}(x) = (\max(0, 1 - x))^2$, logistic loss $\ell_L(x) = \log(1 + e^{-x})$, and exponential loss $\ell_E(x) = e^{-x}$.

In the primal problem in Eq. (5), the computation of the basis function matrix $\Phi(\mathbf{x}_i)$ may be infeasible, or otherwise heuristically approximated by some kernel matrix factorization, as the basis function $\phi(x_i)$ could be a mapping to infinite dimensions for many kernels. To avoid such dilemma, we first derive the dual problem, then introduce the algorithm for recovering the weight from the dual solution. An illustration of the dual framework based on the primal framework is demonstrated in Figure 1. For the theorems in this section, we provide proof sketches and leave the detailed derivations in Appendix A.

### 3.1 Dual Optimization Problem

In order to avoid the ambiguous approximation from kernel matrix factorization, we consider the dual form of problem (5) and compute a kernel generating matrix directly with the benefits of the kernel trick (Schölkopf et al., 2002).

**Theorem 1.** *The dual problem of Eq. (5) is given by:*

$$\underset{\alpha}{\text{maximize}} \quad -c\sum_{i=1}^{n} \ell^{*}\left(-\frac{\alpha_i}{c}\right) \tag{6}$$

$$\text{subject to} \quad \lambda_{\max}\Big(\sum_{i=1}^{n}\sum_{j=1}^{n}\alpha_i\alpha_j y_i y_j K(\mathbf{x}_i, \mathbf{x}_j)\Big) \le 1\,,$$

$$\alpha_i \ge 0\,, \quad \forall i \in [n]\,,$$

*in which $\alpha_i$'s are the dual variables, $\ell^{*}(\cdot)$[1] is the Fenchel conjugate of the loss function $\ell(\cdot)$, $K(\mathbf{x}_i, \mathbf{x}_j) = \Phi(\mathbf{x}_i)^{\top}\Phi(\mathbf{x}_j)$ is a kernel generating matrix, such that its entries are composed of the kernel function $\mathcal{K}(\cdot,\cdot) = \phi(\cdot)^{\top}\phi(\cdot)$ taking one patch from each of the two samples as input:*

$$K\left(\mathbf{x}_i, \mathbf{x}_j\right) = \begin{pmatrix} \mathcal{K}\big(z_1(\mathbf{x}_i), z_1(\mathbf{x}_j)\big) & \cdots & \mathcal{K}\big(z_1(\mathbf{x}_i), z_p(\mathbf{x}_j)\big) \\ \vdots & \ddots & \vdots \\ \mathcal{K}\big(z_p(\mathbf{x}_i), z_1(\mathbf{x}_j)\big) & \cdots & \mathcal{K}\big(z_p(\mathbf{x}_i), z_p(\mathbf{x}_j)\big) \end{pmatrix}. \tag{7}$$

*Proof Sketch:* The derivation of the dual problem follows the standard Lagrangian duality framework, with a careful construction of the Fenchel conjugate (Boyd & Vandenberghe, 2004) for the nuclear norm and the loss function. For Lagrangian variables $\alpha_i \ge 0$, we can write the Lagrangian function of the primal problem in Eq. (5) as follows, after rearranging terms and utilizing the trace definition of matrix inner product:

$$\mathcal{L}\left(A, \xi, \alpha\right) = \|A\|_{*} - \Big\langle \sum_{i=1}^{n}\alpha_i y_i \Phi\left(\mathbf{x}_i\right), A\Big\rangle + c\sum_{i=1}^{n}\ell(\xi_i) + \sum_{i=1}^{n}\alpha_i\xi_i. \tag{8}$$

Then by definition, we get the dual function by minimizing the Lagrangian function with respect to the primal variables:

$$g\left(\alpha\right) = \min_{A,\xi}\mathcal{L}\left(A, \xi, \alpha\right) = \underbrace{-\max_{A}\Big\langle \sum_{i=1}^{n}\alpha_i y_i \Phi\left(\mathbf{x}_i\right), A\Big\rangle - \|A\|_{*}}_{g_1} \underbrace{-\max_{\xi} -c\sum_{i=1}^{n}\ell(\xi_i) - \sum_{i=1}^{n}\alpha_i\xi_i}_{g_2}, \tag{9}$$

in which $g_1$ and $g_2$ take the form of the Fenchel conjugate. The conjugate function of norms is their dual norm (Boyd & Vandenberghe, 2004), then for the norm $f_0 = \|A\|_{*}$, we have

$$g_1 = f_0^{*}\Big(\sum_{i=1}^{n}\alpha_i y_i \Phi\left(\mathbf{x}_i\right)\Big),$$

$$\text{in which } f_0^{*}\left(\cdot\right) = \begin{cases} 0\,, & \|\cdot\|_2 \le 1 \\ \infty, & otherwise \end{cases}. \tag{10}$$

Also,

$$g_2 = c\sum_{i=1}^{n}\ell^{*}\left(-\frac{\alpha_i}{c}\right),$$

---

[1]To have a detailed illustration of $\ell^{*}(\cdot)$, we include the Fenchel conjugate of some common losses in Appendix C.

in which $\ell^*(\cdot) = \sup_{\xi_i} \left\{ \langle \cdot, \xi_i \rangle - \ell(\xi_i) \right\}$ is the conjugate function of the loss function $\ell(\xi_i)$. Therefore, the dual problem of Eq. (5) is as follows:

$$\begin{aligned} \operatorname*{maximize}_{\alpha} \quad & -c \sum_{i=1}^{n} \ell^* \left( -\frac{\alpha_i}{c} \right) \\ \text{subject to} \quad & \big\| \sum_{i=1}^{n} \alpha_i y_i \Phi\left(\mathbf{x}_i\right) \big\|_2 \leq 1 \,, \\ & \alpha_i \geq 0 \,, \quad \forall i \in [n] \,. \end{aligned} \tag{11}$$

Finally, we construct the kernel function by squaring the spectral norm constraint in Eq. (11) on both sides, which leads to the form in Eq. (6). ∎

The dual problem presents in the form of the kernel generating matrix and eliminates the basis function, and consequently the ambiguity of factorizing the kernel matrix. Also, our kernel generating matrix is of size $\mathcal{O}(p^2)$, which is more computationally efficient in terms of spatial complexity.

With the dual problem constructed, we solve the dual problem by a coordinate-descent algorithm. The algorithm picks the coordinate entry in ascending order with respect to the maximum eigenvalues of the kernel generating matrices, and optimize the chosen coordinate by binary search. The pseudocode of the algorithm is provided in Algorithm 2 in Appendix E.1.

## 3.2 Recovering the Parameters

In conventional machine learning dual optimization problems like the dual of Support Vector Machine (SVM), the primal solution can be exactly recovered with the stationary condition in the Karush–Kuhn–Tucker (KKT) conditions (Weston & Watkins, 1998; Vapnik, 1998). Such closed form expression for the primal solution, however, no longer applies to the DCCNN formulation, because the nuclear norm is non-differentiable. To tackle this problem, we propose a parameter recover algorithm that avoids recovering the primal solution $\hat{A}$, but directly recovers the linear weight $\hat{L}$ and the output of the convolutional layer $\Phi(x)^\top \hat{W}$ without explicitly recovering $\hat{W}$ or computing $\Phi(x)$. Our algorithm leverages the KKT conditions and the subdifferential set of the nuclear norm.

Since there are infinite many ways to decompose the convolutional weight $\hat{W}$ and linear weight $\hat{L}$ from the optimal parameter $\hat{A}$, we adopt the singular value decomposition method proposed by Zhang et al. (2017), i.e., for the compact SVD $\hat{A} = \hat{U}_1 \hat{D}_1 \hat{V}_1$, regard $\hat{U}_1$ as the convolutional weight $\hat{W}$ and $\hat{V}_1$ as the linear weight $\hat{L}$. We use them interchangeably in the following context. For recovering such weight parameters, we propose an algorithm to recover the linear weight in Theorem 2, and we propose the method to implicitly recover the convolutional weight by the convolution output in Theorem 3. The complete algorithm workflow is demonstrated in Algorithm 1.

### 3.2.1 Recovering the Linear Weight

Our weight recovering algorithm leverages the KKT conditions, in which the subdifferential of the nuclear norm plays an important part. We begin by introducing two Lemmas that formalize the stationary condition of our optimization problem and the relevant definition for the nuclear norm subdifferential.

**Lemma 1.** *For the primal problem* (5) *and the dual variables* $\{\alpha_i\}_{i=1}^n$*, the stationary condition with respect to variable $A$ in the KKT conditions is given by:*

$$0 \in \partial \|\hat{A}\|_* - \sum_{i=1}^{n} \hat{\alpha}_i y_i \Phi\left(\mathbf{x}_i\right). \tag{12}$$

**Lemma 2** (Subdifferential of Nuclear Norm (Watson, 1992))**.** *For matrix $A \in \mathbb{R}^{d_2 \times p}$ with* $\operatorname{rank}(A) = r$*, consider its compact SVD $A = U_1 D_1 V_1^\top$ in which $U_1 \in \mathbb{R}^{d_2 \times r}$, $D_1 \in \mathbb{R}^{r \times r}$, and $V_1 \in \mathbb{R}^{p \times r}$, and full SVD $A = UDV^\top$ in which $U \in \mathbb{R}^{d_2 \times d_2}$, $D \in \mathbb{R}^{d_2 \times p}$, and $V \in \mathbb{R}^{p \times p}$. Denote $U_2 \in \mathbb{R}^{d_2 \times (d_2 - r)}$, $V_2 \in \mathbb{R}^{p \times (p-r)}$ such*

---

**Algorithm 1** Recovering the weight parameters for two-layer DCCNNs

---

**Input:** Data $\{(\mathbf{x}_i, y_i)\}_{i=1}^n$; Optimized dual solution $\{\hat{\alpha}_i\}_{i=1}^n$ to problem (6); Kernel function $\mathcal{K}$.
**Recover the Linear Weight:**

1: Compute $S = \sum_{i,j} \hat{\alpha}_i \hat{\alpha}_j y_i y_j K(\mathbf{x}_i, \mathbf{x}_j)$. {Using $K(\cdot, \cdot)$ defined with $\mathcal{K}$ in Eq. (7) without ambiguous factorization}
2: Compute the eigendecomposition $S = \tilde{V}\Lambda\tilde{V}^{-1}$.
3: Let the eigenvectors with eigenvalue 1 form the linear weight $\hat{L}$, i.e.,

$$\hat{L} = \left[\tilde{V}(i)\right] \in \mathbb{R}^{p \times r}, \text{ for all } i \text{ such that } \Lambda_{ii} = 1.$$

**Recover the Convolution Output (Input for the next layerwise training):** {Without computing $\Phi(\mathbf{x}_i)$ or $W$}

4: Compute the output of the convolutional layer $\Phi(\mathbf{x}_i)^\top \hat{W}$ by

$$\Phi(\mathbf{x}_i)^\top \hat{W} = \sum_{j=1}^n \hat{\alpha}_j y_j K(\mathbf{x}_i, \mathbf{x}_j)\hat{L}.$$

**Output:** Linear weight $\hat{L}$; Output of convolutional layer $\{\Phi(\mathbf{x}_i)^\top \hat{W}\}_{i=1}^n$.

---

*that $U = [U_1 \mid U_2]$, $V = [V_1 \mid V_2]$. Then the subdifferential set of $A$ is given by*

$$\partial \|A\|_* = \left\{ U_1 V_1^\top + U_2 E V_2^\top : E \in \mathbb{R}^{(d_2-r)\times(p-r)}, \ \sigma_{\max}(E) \leq 1 \right\}, \tag{13}$$

*in which $\sigma_{\max}(\cdot)$ is the largest singular value.*

Now we introduce our method of recovering the linear weight using the dual solution and the kernel generating matrix, and validate that what we recover is indeed the linear weight $\hat{V}_1$.[2]

**Theorem 2.** *Given the optimal dual solution $\{\hat{\alpha}_i\}_{i=1}^n$ to the problem in Eq. (6), let $\tilde{V} \in \mathbb{R}^{p \times p}$ be the matrix of eigenvectors from the eigendecomposition $\tilde{V}\Lambda\tilde{V}^{-1} = \sum_{i=1}^n \sum_{j=1}^n \hat{\alpha}_i \hat{\alpha}_j y_i y_j K(\mathbf{x}_i, \mathbf{x}_j)$, where $\Lambda = \operatorname{diag}(\lambda_1, ..., \lambda_p) \in \mathbb{R}^{p \times p}$. The linear weight $\hat{V}_1$ can be recovered by the eigenvectors in $\tilde{V}$ corresponding to the eigenvalue of 1, that is,*

$$\hat{V}_1 = \left[\tilde{V}(i)\right] \in \mathbb{R}^{p \times r}, \text{ for all } i \text{ such that } \lambda_i = 1, \tag{14}$$

*in which $\tilde{V}(i)$ denotes the $i^{th}$ column of $\tilde{V}$ and $[\cdot]$ denotes the concatenation of these columns.*

*Proof Sketch.* By the stationary condition in Lemma 1, we know that the there exists one element in the subdifferential of $\|A\|$ at value $\hat{A}$ that satisfies $\sum_{i=1}^n \hat{\alpha}_i y_i \Phi(\mathbf{x}_i) \in \partial\|\hat{A}\|_*$. Then for $\hat{A}$, consider its compact SVD $\hat{A} = \hat{U}_1 \hat{D}_1 \hat{V}_1^\top$ and the full SVD $\hat{A} = \hat{U}\hat{D}\hat{V}^\top$ in which $\hat{U} = [\hat{U}_1 \mid \hat{U}_2]$, $\hat{V} = [\hat{V}_1 \mid \hat{V}_2]$. By Lemma 2, we know that there exists $\hat{E}$ such that $\hat{U}_1 \hat{V}_1^\top + \hat{U}_2 \hat{E} \hat{V}_2^\top = \sum_{i=1}^n \hat{\alpha}_i y_i \Phi(\mathbf{x}_i)$. Now we construct the kernel generating matrix under the intuition to avoid the basis function matrix $\Phi(\mathbf{x}_i)$. Let $T = \hat{U}_1 \hat{V}_1^\top + \hat{U}_2 \hat{E} \hat{V}_2^\top = \sum_{i=1}^n \hat{\alpha}_i y_i \Phi(\mathbf{x}_i)$. By computing $S = T^\top T$ we get the kernel generating matrix $K(\mathbf{x}_i, \mathbf{x}_j)$ on the right side of the equation. For the left side, we know by the properties of SVD that $\hat{U}_1$ and $\hat{U}_2$ are semi-orthogonal matrices, i.e., $\hat{U}_1^\top \hat{U}_1 = \mathbf{I}_r$, $\hat{U}_2^\top \hat{U}_2 = \mathbf{I}_{(d_2-r)}$, and also $\hat{U}_1 \perp \hat{U}_2$, i.e., $\hat{U}_1^\top \hat{U}_2 = \hat{U}_2^\top \hat{U}_1 = \mathbf{0}$. Therefore, by plugging in $T = \hat{U}_1 \hat{V}_1^\top + \hat{U}_2 \hat{E} \hat{V}_2^\top$ and with some simplification, we get $S = \hat{V}_1 \hat{V}_1^\top + \hat{V}_2 \hat{E}^\top \hat{E} \hat{V}_2^\top$. Also, $\hat{V}_1 \perp \hat{V}_2$, then $\hat{V}_1 \perp \hat{V}_2 \hat{E}^\top$. Consequently, there exists $\tilde{V} \in \mathbb{R}^{p \times p}$ for the eigendecompositions $S = \tilde{V}\Lambda\tilde{V}^{-1}$, $\hat{V}_1 \hat{V}_1^\top = \tilde{V}\Lambda_v \tilde{V}^{-1}$ and $\hat{V}_2 \hat{E}^\top \hat{E} \hat{V}_2^\top = \tilde{V}\Lambda_e \tilde{V}^{-1}$, such that $\Lambda = \Lambda_v + \Lambda_e$. Since $\hat{V}_1$ is semi-orthogonal, there are only 0's and 1's on the diagonal of $\Lambda_v$. Meanwhile, by the Lemma 2, we know that $\sigma_{\max}(E) \leq 1$. For the moment, assume that $\sigma_{\max}(E) < 1$. That is to say, the elements on the diagonal of $\Lambda_e$ lie in the range $[0, 1)$. Moreover, given that $\hat{V}_1 \perp \hat{V}_2 \hat{E}^\top$, we know that $\Lambda_{v,ii} = 0$ for all $i$ with $\Lambda_{e,ii} > 0$, and $\Lambda_{v,ii} = 1$ otherwise. Therefore, the eigenvectors $\left[\tilde{V}(i)\right]$ with $\Lambda_{ii} = 1$ are the eigenvectors of $\hat{V}_1 \hat{V}_1^\top$. Since $\hat{V}_1$ is from the compact SVD with $\operatorname{rank}(\hat{V}_1) = r$, $\hat{V}_1 = \left[\tilde{V}(i)\right]$ is the recovered linear weight. Finally, recall that in general $\sigma_{\max}(E) \leq 1$.

---

[2]Strictly speaking, we are recovering a solution in a superset of the set of vectors in the linear weight.

Therefore, the eigenvalues of $\hat{V}_2 \hat{E}^\top \hat{E} \hat{V}_2^\top$ could also be 1, and thus, we are recovering a solution in a superset of the set of vectors in the linear weight.[3] $\qquad\square$

**Remark 1.** *One of the benefits we enjoy from our recovery algorithm is that we do not need to set the number of filters as a hyperparameter, but can directly deduce it from the linear weight we recover. That is, we do not fix the number of convolutional filters $r$, but deduce $r$ by letting $r := \mathrm{rank}(\hat{A})$, or equivalently, $r := \mathrm{rank}(\hat{V}_1)$, i.e., $r$ is the number of 1's in the diagonal of $\Lambda$. Furthermore, the imposed nuclear norm constraint on $A$ encourages a low rank which implicitly reduces the parameter size in attempt to find the minimum number of filters necessary.*

### 3.2.2 Recovering the convolutional weight

In the case of the convolutional weight, there is no closed form expression for $\hat{U}_1 \in \mathbb{R}^{d_2 \times r}$ because we cannot exactly compute $\Phi(\mathbf{x}_i) \in \mathbb{R}^{d_2 \times p}$ and $d_2$ could be infinity. We provide an approach to compute the output of the convolutional weight directly, which implicitly contains $\hat{U}_1$, and can be used for the purpose of layerwise training as well as making new predictions.

**Theorem 3.** *Given the optimal dual solution $\hat{\alpha}_i$, $i \in [n]$ to the problem in Eq. (11), the convolutional weight $\hat{U}_1$ can be implicitly recovered by the convolution output, that is, $\forall i \in [n]$,*

$$\Phi(\mathbf{x}_i)^\top \hat{U}_1 = \sum_{j=1}^n \hat{\alpha}_j y_j K(\mathbf{x}_i, \mathbf{x}_j) \hat{V}_1. \tag{15}$$

*Proof Sketch.* By Lemma 1 and Lemma 2 we know that $\hat{V}_1 \hat{U}_1^\top = \sum_{j=1}^n \hat{\alpha}_j y_j \Phi(\mathbf{x}_j)^\top - \hat{V}_2 \hat{E}^\top \hat{U}_2^\top$. For both sides of the equation, multiply with $\hat{V}_1^\top$ on the left and $\Phi(\mathbf{x}_i)$ on the right, we have $\hat{V}_1^\top \hat{V}_1 \hat{U}_1^\top \Phi(\mathbf{x}_i) = \sum_{j=1}^n \hat{\alpha}_j y_j \hat{V}_1^\top \Phi(\mathbf{x}_j)^\top \Phi(\mathbf{x}_i) - \hat{V}_1^\top \hat{V}_2 \hat{E}^\top \hat{U}_2^\top \Phi(\mathbf{x}_i)$. Note that $\hat{V}_1$ is semi-orthogonal and $\hat{V}_1 \perp \hat{V}_2$, thus $\hat{V}_1^\top \hat{V}_1 = \mathbf{I}$ and $\hat{V}_1^\top \hat{V}_2 = \mathbf{0}$. This concludes the proof. $\qquad\square$

After recovering the convolution output, one can train a multi-layer DCCNN by the layerwise training approach applied in (Zhang et al., 2017; Belilovsky et al., 2019). That is to regard the vectorized output of the convolution weight $\mathrm{vec}\left(\Phi(\mathbf{x}_i)^\top \hat{U}_1\right)$ as the input for the next layer. Note that the vectorization automatically takes care of the case for multiple-channel input or output, as it will eventually be transformed into a vector. The complete algorithm of learning a $\mathcal{D}$-layer DCCNN is illustrated in Algorithm 3 in Appendix E.2.

Our way of recovering the weight output can also be adapted to other CNN techniques such as *Average Pooling.* To achieve average pooling, one can generate a pooling matrix $G$ and simply multiply it with the convolution output before feeding it into the next layerwise training. We refer the reader to Appendix F for the generation of average pooling matrix.

Making prediction with a new sample $\mathbf{x}_{new}$ can also be achieved with the dual solution and the kernel generating matrix. We can make predictions with the two-layer DCCNN by firstly plugging the sample into Eq. (15), then multiplying the convolutional output with the linear weight, and finally computing $f(\mathbf{x}_{new}) = \mathrm{sign}\left(\mathrm{Tr}(\sum_{j=1}^n \hat{\alpha}_j y_j K(\mathbf{x}_{new}, \mathbf{x}_j) \hat{L} \hat{L}^\top)\right)$. The algorithm for making predictions with a layerwise trained $\mathcal{D}$-layer DCCNN is described in Algorithm 4 in Appendix E.3.

### 3.3 Extension to Multiclass Classification

Our proposed method can be easily extended to multi-class classification. In this section, we provide results for the multi-class version and leave the derivations and proofs in Appendix B, as they closely resemble the derivations for binary classification.

We start from the multiclass version of loss minimization in Eq. (4). To be consistent with the binary classification formulation, for the parameters $\{A_k\}_{k=1}^m$, we define $A = [A_1, ..., A_m] \in \mathbb{R}^{d_2 \times mp}$. Similarly, for $\Phi(\mathbf{x}_i) \in \mathbb{R}^{d_2 \times p}$, we define $\Phi_k'(\mathbf{x}_i) = \left[\mathbf{0}_{d_2 \times (k-1)p}, \Phi(\mathbf{x}_i), \mathbf{0}_{d_2 \times (m-k)p}\right] \in \mathbb{R}^{d_2 \times mp}$. In this way we have

---

[3]Experimentally, the total number of vectors associated to eigenvalues close to 1 was very small. Thus, most likely we are recovering only $\hat{V}_1$ and not $\hat{V}_2$, and so we do not believe this is an issue in practical terms.

| Dataset | Architecture | CCNN | | | | SGD (end-to-end / layerwise) | DCCNN |
|---------|--------------|------|------|------|----------|------|-------|
| | | $UD^{\frac{1}{2}}$ | $UD^{\frac{1}{2}}V$ | $K^{\frac{1}{2}}$ | Cholesky | | |
| MNIST | 1-Conv-Layer | 90.3% | 90.3% | 88.8% | 93.7% | 90.2% / —— | 94.8% |
| | 2-Conv-Layer | 93.7% | 93.5% | 93.5% | 96.7% | 95.2% / 92.4% | 96.0% |
| ImageNet | AlexNet | 62.3% | 62.3% | 53.2% | 62.0% | 87.1% / 86.3% | 83.3% |
| | VGG11 | 55.3% | 52.5% | 59.8% | 57.0% | 89.2% / 86.7% | 85.0% |

(a) Test Accuracies on MNIST and ImageNet Binary Classification.

| Dataset | Architecture | CCNN | | | | SGD | DCCNN |
|---------|--------------|------|------|------|----------|------|-------|
| | | $UD^{\frac{1}{2}}$ | $UD^{\frac{1}{2}}V$ | $K^{\frac{1}{2}}$ | Cholesky | | |
| MNIST | 1-Conv-Layer | 75.4% | 75.4% | 82.3% | 85.9% | 87.0% | 85.3% |

(b) Test Accuracies on MNIST Multiclass Classification.

Table 1: Experiment results for binary and multiclass classification. Methods compared include CCNN, CNN trained with SGD, and our proposed DCCNN. For CCNN we use a 25-column matrix $Q$ from 4 different ways of kernel matrix factorization: (1) $UD^{\frac{1}{2}}$ for $K = UDU^{\top}$; (2) $UD^{\frac{1}{2}}V$ for $K = UDU^{\top}$, and random orthonormal matrix $V$; (3) $K^{\frac{1}{2}}$; (4) the Cholesky decomposition of $K$. The X-Conv-Layer in the architecture column refers to the number of convolutional layers, as only one linear layer is concatenated at the end for classification. For architectures with more than one convolutional layer, we list the result of SGD trained both end-to-end and layerwise. The result of SGD only reflects its performance in our specific experiment setting, e.g. few convolutional filters, and is listed for DCCNN sanity check purposes.

$\text{Tr}\left(\Phi(\mathbf{x}_i)^{\top}A_k\right) = \text{Tr}\left(\Phi'_k(\mathbf{x}_i)^{\top}A\right)$, and the minimization in Eq. (4) is equivalent to the following optimization problem:

$$\underset{A}{\text{minimize}} \quad \|A\|_* + c\sum_{k \neq y_i}\sum_{i=1}^{n}\ell(\xi_{k,i}) \tag{16}$$

$$\text{subject to} \quad \text{Tr}\left(\Phi'_{y_i}(\mathbf{x}_i)^{\top}A\right) - \text{Tr}\left(\Phi'_k(\mathbf{x}_i)^{\top}A\right) \geq \xi_{k,i}, \quad \forall i \in [n], \ k \neq y_i$$

in which $\ell(\cdot)$ is convex and non-increasing, $\xi_{k,i}$ is an equivalent variable of $\text{Tr}\left(\Phi'_{y_i}(\mathbf{x}_i)^{\top}A\right) - \text{Tr}\left(\Phi'_k(\mathbf{x}_i)^{\top}A\right)$, and $c > 0$ is some hyperparameter.

We can derive the dual of the optimization problem in Eq. (16) as in Section 3.1. With an adaptation of the kernel generating matrix, we give out the dual problem for multi-class classification.

**Theorem 4.** *For all dual variables $\alpha_{k,i} \geq 0$ with $i \in [n]$ and $k \in [m]$, the dual problem of Eq. (16) is given by:*

$$\underset{\alpha}{\text{maximize}} \quad -c\sum_{i=1}^{n}\sum_{k=1}^{m}\ell^*\left(-\frac{\alpha_{k,i}}{c}\right) \tag{17}$$

$$\text{subject to} \quad \lambda_{\max}\left(\sum_{i=1}^{n}\sum_{j=1}^{n}\sum_{k=1}^{m}\alpha'_{k,i}\alpha'_{k,j}K'_k(\mathbf{x}_i, \mathbf{x}_j)\right) \leq 1,$$

$$\alpha_{k,i} \geq 0, \ \alpha_{y_i,i} = 0, \ \forall i \in [n], \ \forall k \in [m],$$

*in which $\alpha'_{k,i} = \sum_{s=1}^{m}\alpha_{s,i}\mathbb{1}[k = y_i] - \alpha_{k,i}$, $\ell^*(\cdot)$ is the Fenchel conjugate of the loss function $\ell(\cdot)$, and the kernel generating matrix is constructed in $K'_k(\mathbf{x}_i, \mathbf{x}_j) = \text{diag}\left(\mathbf{0}_{p \times p}, ..., \underbrace{K(\mathbf{x}_i, \mathbf{x}_j)}_{k^{th}\ block\ diagonal}, ..., \mathbf{0}_{p \times p}\right)$, with*

$K(\mathbf{x}_i, \mathbf{x}_j) = \Phi(\mathbf{x}_i)^{\top}\Phi(\mathbf{x}_j)$.

After solving the dual problem, we can recover the linear weight and the convolution output by Theorem 5[4] and Theorem 6 using the dual solution $\{\{\hat{\alpha}_{k,i}\}_{k=1}^m\}_{i=1}^n$ and the kernel generating matrix.

**Theorem 5.** *We can recover the linear weight $\hat{V}_1 \in \mathbb{R}^{mp \times r}$ for multi-class classification by $\hat{V}_1 = \left[\tilde{V}(i)\right]$, for all $i$ with $\lambda_i = 1$. Here $\tilde{V} \in \mathbb{R}^{mp \times mp}$ comes from $\tilde{V}\Lambda\tilde{V}^{-1} = \sum_{i=1}^n \sum_{j=1}^n \sum_{k=1}^m \alpha'_{k,i}\alpha'_{k,j} K'_k\left(\mathbf{x}_i, \mathbf{x}_j\right)$ where $\Lambda = \mathrm{diag}(\lambda_1, ..., \lambda_{mp}) \in \mathbb{R}^{mp \times mp}$, and $\alpha'_{k,i} = \sum_{s=1}^m \alpha_{s,i}\mathbb{1}\left[k = y_i\right] - \alpha_{k,i}$.*

**Theorem 6.** *We can implicitly recover the convolutional weight in the convolution output for multi-class classification. That is, for all $i \in [n]$, we have $\Phi\left(\mathbf{x}_i\right)^\top \hat{U}_1 = \sum_{j=1}^n \sum_{k=1}^m \alpha'_{k,j}\left[\mathbf{0}_{p \times (k-1)p}, K\left(\mathbf{x}_i, \mathbf{x}_j\right), \mathbf{0}_{p \times (m-k)p}\right]\hat{V}_1$.*

Same as binary classification, $\mathrm{vec}\left(\Phi\left(\mathbf{x}_i\right)^\top \hat{U}_1\right)$ can be regarded as the input for the next layerwise training. To make a new prediction for sample $\mathbf{x}_{new}$ with the multiclass model, for every $k \in [m]$, we can extract the linear weight of each class $\hat{L}_k \in \mathbb{R}^{p \times r}$ from $\hat{L} = [\hat{L}_1^\top, ..., \hat{L}_m^\top]^\top \in \mathbb{R}^{mp \times r}$, and then make predictions with the multi-class predictor $f(\mathbf{x}_{new}) = \arg\max_{k \in [m]} \left\{ \mathrm{Tr}\left(\sum_{i=1}^n \sum_{l=1}^m \alpha'_{l,i}\left[\mathbf{0}_{p \times (l-1)p}, K(\mathbf{x}_{new}, \mathbf{x}_i), \mathbf{0}_{p \times (m-l)p}\right]\hat{L}\hat{L}_k^\top\right)\right\}$.

## 4 Experiments

In this section, we evaluate DCCNN on real-world data as a sanity check for the proposed method. The performance is by no means state-of-the-art for relevant tasks, and the baselines reflect only its performance in our experimental settings, which is the same for all methods. For DCCNN, we solve the dual problem by a coordinate-descent approach (Algorithm 2 in Appendix E.1). For CCNN, we used a projected gradient descent approach and $r = 25$. We apply hinge loss for evaluation. The specific form of the problem in Eq. (6) with hinge loss can be found in Appendix D. We briefly report results in Table 2, and leave the detailed analysis in Appendix G. In Table 2a, for the binary classification task of the MNIST data (Lecun et al., 1998), we can see the DCCNN outperforms CNNs trained using SGD and different kernel matrix factorizations for CCNN on one-conv-layer and two-conv-layer networks with only one exception, which verifies the effectiveness of DCCNN. On the more complicated ImageNet dataset (Deng et al., 2009), DCCNN also performs comparably well with the end-to-end SGD optimized CNNs under both AlexNet (Krizhevsky et al., 2012) and VGG11 (Simonyan & Zisserman, 2015) architectures, and significantly outperforms the CCNN method. From Table 2b, we see that in multiclass classification, the performance level of DCCNN is better than various factorization versions of CCNN and is comparable with SGD.

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
