# OpenReview forum: "On the Dual Problem of Convexified Convolutional Neural Networks"
_TMLR — Accepted by TMLR_

### Review · Reviewer_1tmZ · 2023-09-18

**Summary Of Contributions:**

This paper studies the problem of solving Convexified Convolutional Neural Networks by their dual problem, the DCCNN. The paper introduces an equivalent problem for the CCNN, which can be dualized. The authors explore the computational benefits of the dual counterpart and show how to solve it and recover the primal solution. The authors provide numerical experiments that show that their method is competitive with existing methods. The paper presents a clever novel idea that leads to a more efficient way of obtaining CCNNs.

**Audience:**

Yes

**Broader Impact Concerns:**

None.

**Claims And Evidence:**

Yes

**Requested Changes:**

## Minor comments

1. In the appendix, page 16, proof of Theorem 1, in the first line, $\beta_i$ are introduced but never used. In the same proof, the proof
2. Algorithm 2 in the appendix needs some rewriting. For example: add a “for all i” to line 2. Lines 11 and 14 are the same, they should be taken out of the “if”.
3. In the experiment setting, could the authors provide an intuitive explanation on why the variance of their method is at least 3 times larger than SGD. Is this because of the initialization? “For the network structure similar to AlexNet on the ImageNet dataset, we observe a standard deviation of 2.5%, and for the architecture similar to VGG11, the standard deviation is 1.7%. For multiclass classification, the standard deviation of SGD is 0.5%. “
4. The authors’ method considers a two-layer CNN for the theoretical results. However, in the experiments, they use AlexNet DCCNN, and VGG11 DCCNN. Given that AlexNet and VGG are not two-layer CNNs, I would suggest adding an explanation of how these are obtained.
5. The authors claim that their method is “Our DCCNN framework is highly efficient. It operates without loading the full kernel matrix for all
samples in memory or introducing any unnecessary memory overhead of factorization. As a result,
DCCNN inherits all the statistical benefits of convexified CNNs, while enjoying a much more efficient
workflow.” without providing numerical evidence or citations. The reader needs to go as far as Appendix G4 to find the proper comparison. I would suggest adding the values, and a reference to this section in the main body of the paper.

**Strengths And Weaknesses:**

## Major Strengths

1. Solving the dual problem (6) is a novel idea that, as the authors point out has a number of computational benefits. This idea is possible by the introduction of the slackness $\xi_i$  in problem (5) which is also clever. In all, this paper presents a novel and useful idea.
2. The paper is very well written, and the concepts are introduced in a smooth way. The paper’s claims are correctly supported by both theoretical results and experiments.
3. The proofs are correct and well-written. The appendix is easy to follow, and the notation is clear and unambiguous.
4. The authors provide an extensive amount of analysis of their method. They analyze the computation required to run their method, and they provide training times. I would also suggest making their code available.

## Major Weaknesses

1. The authors claim that “The whole convexified approach allows for efficient and optimal training of the network, and from a theoretical point of view makes precise statistical analysis viable (Zhang et al., 2017)." However, their method is an order of magnitude slower than SGD (table 2 page 28). The authors should soften this claim.
2. The experiments show that the author's method is in the ballpark of state-of-the-art results. However, I believe that the details of the experimental setup should be given in the main body of the paper. The authors claim in the main body of the paper that their experiments were carried out in MNIST and ImageNet, but this is not entirely true:

“**Data.** We use the MNIST (Lecun et al., 1998) and the ImageNet datasets (Deng et al., 2009) under the
terms of the Creative Commons Attribution-Share Alike 3.0 license. In binary classification experiments,
for both the MNIST and ImageNet datasets, we randomly pick two classes, images of digit 2 versus 3 from
MNIST, images containing tench versus bathroom tissue from ImageNet, and use 2000 images for training,
100 images for validation, and 600 images for testing. We center crop the ImageNet images to [224,224] and
transfer the pixel value to [0, 1]. No other preprocessing is applied.
”

No details on the multi-class classification are provided. How many classes do they consider? How many samples per class?

A related question is, when the authors say that “We center crop the ImageNet images to [224,224] and
transfer the pixel value to [0, 1]. “ does this mean that the authors consider black and white images?

3. A major limitation of these methods is the fact that they do not work on large datasets given that their complexity scales with the number of samples. The authors should provide a better explanation of why this work is relevant in a context where datasets have an increasing number of samples.

---

> ### Author Response · Authors · 2023-10-21
>
> > ... In all, this paper presents a novel and useful idea.
> >
> > The paper is very well written, and the concepts are introduced in a smooth way. The paper's claims are correctly supported by both theoretical results and experiments.
> >
> > The proofs are correct and well-written. The appendix is easy to follow, and the notation is clear and unambiguous.
> >
> > The authors provide an extensive amount of analysis of their method... I would also suggest making their code available.
>
> We thank the reviewer for the appreciation of our work. We will gladly make the code available.
>
> # Weaknesses
>
> > 1. The authors claim that "The whole convexified approach allows for efficient and optimal training of the network, and from a theoretical point of view makes precise statistical analysis viable (Zhang et al., 2017)." However, their method is an order of magnitude slower than SGD (table 2 page 28)...
>
> We will soften the claim regarding Zhang's CCNN, by mentioning SGD.
>
> > 2. The experiments show that the author's method is in the ballpark of state-of-the-art results. However, I believe that the details of the experimental setup should be given in the main body of the paper.
>
> We will move some of the experimental details from the appendix to the main text.
>
> > No details on the multi-class classification are provided.
>
> The multi-class experiments in Table 2 in Page 27 (Experiment results for binary and multiclass classification) relates to MNIST, in which we used all classes, but limited the number of training images to 2000. This is what we meant by "2000 images for training". We will clarify this.
>
> > when the authors say that "We center crop the ImageNet images to [224,224] and transfer the pixel value to [0, 1]. " does this mean that the authors consider black and white images?
>
> No, we transfer from $[0,255]$ to $[0,1]$ independently for each of the three RGB channels.
>
> > 3. A major limitation of these methods is the fact that they do not work on large datasets given that their complexity scales with the number of samples. The authors should provide a better explanation of why this work is relevant in a context where datasets have an increasing number of samples.
>
> While we agree that this is a very good point, we argue that this might not be able to be accomplished in a single paper. We believe a first paper should motivate that the dual problem in eq.(6) is important (which is part of what we do). Then, another paper would hopefully provide a large-scale algorithm (provided that a first paper has demonstrated that the dual problem is important). For instance, the history of SGD in deep learning is comprised of a series of papers, instead of just one.
>
> # Minor comments
>
> > 1. In the appendix, page 16, proof of Theorem 1, in the first line, $\beta_i$ are introduced but never used.
>
> We will remove $\beta_i$ from the proof. This is related to an older (less general) version of our manuscript. We are sorry for the confusion this might have caused.
>
> > 3. In the experiment setting, could the authors provide an intuitive explanation on why the variance of their method is at least 3 times larger than SGD.
>
> Our paragraph regarding standard deviation relates only to SGD since it has a randomized initialization. We mention "For SGD experiments, in particular, we run 10 trials for each setting with different random seeds and report the average and standard deviation of accuracy. The average accuracy is shown in Table 2. For binary classification, we observe a 1.8% standard deviation for one-layer CNN trained with SGD on the MNIST dataset, and 0.6% for the two-layer CNN trained end-to-end with SGD. For the network structure similar to AlexNet on the ImageNet dataset, we observe a standard deviation of 2.5%, and for the architecture similar to VGG11, the standard deviation is 1.7%. For multiclass classification, the standard deviation of SGD is 0.5%."
>
> > 4. The authors' method considers a two-layer CNN for the theoretical results. However, in the experiments, they use AlexNet DCCNN, and VGG11 DCCNN. Given that AlexNet and VGG are not two-layer CNNs, I would suggest adding an explanation of how these are obtained.
>
> Please see Algorithm 3 in Appendix E.2. In Page 9, we mention "After recovering the convolution output, one can train a multi-layer DCCNN by the layerwise training
> approach applied in (Zhang et al., 2017; Belilovsky et al., 2019)"
>
> > 5. The authors claim ... "... As a result, DCCNN inherits all the statistical benefits of convexified CNNs..." without providing numerical evidence or citations. The reader needs to go as far as Appendix G4 to find the proper comparison. I would suggest adding the values, and a reference to this section in the main body of the paper.
>
> Regarding the statistical benefits CCNNs, we will refer to Theorem 1 in (Zhang et al., 2017). We will make the other proposed minor corrections.

---

> > ### Comment · Reviewer_1tmZ · 2023-10-23
> > **Reply**
> >
> > I appreciate your reply.
> > All of my comments have been addressed.

---

### Review · Reviewer_BXpE · 2023-10-02

**Summary Of Contributions:**

This paper considers the dual problem of Convexified CNNs, dubbed DCCNN. They show that considering DCCNNs has a couple of advantages over CNNs:
+ As an intermediate step for CCNN, one needs to find a kernel matrix factorization. Such factorization is not unique (which, according to the authors, might cause problems). The DCCNN formulation works directly with the kernel matrix.
+ The original CCNN paper obtained a low-rank solution by truncating an SVD. In this paper, the authors propose to include a nuclear norm regularizer to promote low-rankness. Further, they show that the DCCNN can automatically select the solution's rank without needing an additional hyperparameter.
+ They propose an optimization algorithm that doesn't require explicitly forming the kernel matrix, which yields computational gains.

**Audience:**

Yes

**Broader Impact Concerns:**

No concerns.

**Claims And Evidence:**

No

**Requested Changes:**

# Major comments

+ I am not sure I believe that the lack of uniqueness of the factorization $K = LL^\top$ can cause problems. This is highlighted in the introduction as one of the paper's main contributions. However, using the QR factorization, one can easily see that any such factorization satisfies $L = RQ$ for some orthogonal $Q$ and some lower-triangular $R$. Thus, $K = RR^\top$ is the *unique* Cholesky factorization. Thus, any factorization is the same up to an orthogonal change of coordinates. Moreover, the loss function in (3) is invariant under such a change of coordinates (if we simultaneously rotate $\Phi(x_i)$ and $A$).
+ I also do not buy the claim that DCCNNs bypass the need for the rank parameter $r$. The authors make this claim by finding a correspondence between the multiplicity of the eigenvalue $\lambda = 1$ of a generic subdifferential at the solution and the rank of the solution. This correspondence is rather well-known (something similar happens with the subdifferential of the l_1 norm and the sparsity of a solution in compressed sensing). Further, to find their rank estimate, they have to compute an SVD of a dual matrix and evaluate how many singular values are equal to one, which is analogous to computing an SVD of the primal solution (3), and assess how many singular values are equal to one.
+ The claim "Our analysis in the DCCNN framework is novel. To derive the dual training program, the nuclear norm in the primal problem requires a careful construction of the Lagrangian through Fenchel conjugates" is an exaggeration. Nobody has considered the dual problem before in this context, but taking the dual of a convex problem is a relatively standard and well-understood procedure. For instance, Theorem 1 follows as a corollary of Theorem 3.3.5 of Borwein and Lewis (https://carmamaths.org/resources/jon/Preprints/Books/CaNo2/cano2f.pdf).
+ The authors claim that DCCNN is more efficient than CCNN. Their argument relies on the fact that they have an optimization algorithm that doesn't require forming the whole kernel matrix at once but only on smaller $p \times p$ blocks. They include an algorithm to achieve this in the appendix. However, there is no proof of the correctness of their algorithm. Without such a result, the only provably correct way to solve this problem would be to use a vanilla interior point method, which would incur the same complexity as CCNN.
+ The experiments need more details that should be in the body of the paper. For example, which solvers were used to solve the convex problems? What rank was used to truncate the SVD of CCNN? Where can readers find the code to reproduce the experiments? How much time did it take to train both CCNN and DCCNN?

# Minor comments
I use P4P3 to denote "Page 4 Paragraph 3."
+ P1P1 "The performance of CNNs are so" -> "The performance of CNN is so."
+ P1P1 Add citations to "among others".
+ P1P2 Consider explaining what Q is used for. With the current wording, a first-time reader wouldn't understand the relevance of Q.
+ P1P3 "The factorization of the kernel matrix can be quite heuristic." This is a confusing sentence; consider changing it to "The factorization of the kernel matrix is not unique."
+ P2P1 "catastrophic" -> "prohibitive."
+ P3P1 About the comment: "no closed-form expression that can be used to recover the primal solution from the dual solution directly because of the sub-differential of the nuclear norm." It is unclear that the cause it is truly the description of the subdifferential. I would avoid making claims that are not provable.
+ P4P6 "activation function can be contained" -> "activation function are contained."
+ P4P6 "could be a mapping to infinite dimensions" -> "could be a mapping to an infinite dimensional vector space."
+ P6 Theorem 1 No need to include a sketch of the proof in the body of the paper (move it to the appendix if you want to keep it). Recognize that this is a standard result and cite accordingly.
+ P6 Theorem 1 Why do you have a singular value of a scalar in (6)?
+ P9P7 "One can refer to Appendix F" -> "We refer the reader to Appendix F."
+ P9P9 "we provide the results" -> "we provide theoretical results."
+ P11P5 "optimized by SGD" -> "trained using SGD."

**Strengths And Weaknesses:**

# Strengths
+ This paper is well-written and nicely structured. It was relatively easy to read.
+ The theoretical results appear mathematically sound, although I did not check every detail of their proofs.
+ Moreover, the topic is a good match for TMLR.

# Weaknesses
+ Several changes would improve the writing and enhance its clarity.
+ The authors overstate their contributions, as I detailed in the section below. The authors should address these comments before the paper can be accepted for publication.

---

> ### Author Response · Authors · 2023-10-21
>
> > They propose an optimization algorithm that doesn't require explicitly forming the kernel matrix, which yields computational gains.
> >
> > This paper is well-written and nicely structured. It was relatively easy to read.
> >
> > The theoretical results appear mathematically sound...
> >
> > Moreover, the topic is a good match for TMLR.
>
> We thank the reviewer for the appreciation of our work.
>
> > I am not sure I believe that the lack of uniqueness of the factorization $K = L L^\top$ can cause problems.
> >
> > I also do not buy the claim that DCCNNs bypass the need for the rank parameter $r$.
>
> Please see "Motivation" in our global response to all reviewers.
>
> > The claim "Our analysis in the DCCNN framework is novel. To derive the dual training program, the nuclear norm in the primal problem requires a careful construction of the Lagrangian through Fenchel conjugates" is an exaggeration.
> >
> > Minor comment: Theorem 1 No need to include a sketch of the proof in the body of the paper... Recognize that this is a standard result and cite accordingly.
>
> We claim novelty in terms of the CCNN problem, not in terms of the convex optimization field. As in many machine learning papers, tools from the well-known field of convex optimization are used. Moreover, Reviewer 1tmZ appreciates the novelty. Having said that, we can certainly rephrase or tone down our statement.
>
> > The authors claim that DCCNN is more efficient than CCNN... However, there is no proof of the correctness of their algorithm. Without such a result, the only provably correct way to solve this problem would be to use a vanilla interior point method, which would incur the same complexity as CCNN.
>
> Please see "Coordinate Descent" in our global response to all reviewers.
>
> > The experiments need more details that should be in the body of the paper.
> >
> > How much time did it take to train both CCNN and DCCNN?
>
> For our DCCNN convex problem, we use In Algorithm 2 in Appendix E.1. For CCNN we used a projected gradient descent approach, and $r=25$. We will gladly make the code available.
>
> Please see Appendix G.4 (Discussion on Computational Complexity) in Pages 27-28, which also includes Table 2 in Page 28 (Running time ratio between data and models of different scales) of the Appendix.
>
> > Minor comment: About the comment: "no closed-form expression that can be used to recover the primal solution from the dual solution directly because of the sub-differential of the nuclear norm."
>
> Our claim is a comparison between the nuclear norm and the Frobenius norm squared for matrix (or Euclidean norm squared for vectors). In the latter case, the gradient is the matrix itself (or the vector itself) which leads to a closed-form expression. A classical example is support vector machines for classification (for vectors).
>
> > Minor comment: Theorem 1 Why do you have a singular value of a scalar in (6)?
>
> Please see eq.(7). Note that $K(x_i,x_j)$ is a $p \times p$ matrix, and thus, $\sum_{i=1}^n \sum_{j=1}^n \alpha_i \alpha_j K(x_i,x_j)$ is also a $p \times p$ matrix.
>
> We will make the other proposed minor corrections. Finally, we noticed that the reviewer chose "Claims And Evidence: No". Please let us know if there is anything else that we are missing.

---

> > ### Comment · Reviewer_BXpE · 2023-11-26
> > **Reply to authors**
> >
> > I thank the authors for their answers. Here are some points I am still not satisfied with:
> > - To make comparisons fair, the authors should explicitly select different ranks for CCNN, display their results, and also report the rank that DCNNN found.
> > - I am still confused about why the choice of factorization makes a difference; I might be missing something. Let me be a bit more explicit. Consider problem (12) of Zhang et al. (I'll use their notation):
> > $$\min_A \sum L\left(\left( {tr}(Z(x_i) A_1), \dots,  {tr}(Z(x_i) A_d)\right) ; y_i\right) \text{ s.t. } \|A\|^*\leq R$$
> > where $\|A\|^*$ denotes the nuclear norm and $A = (A_1, \dots, A_d)$. Let $A^\star$ be the solution of such a problem, this yields a predictor of the form $$f(x) = (\left( \text{tr}(Z(x) A_1), \dots,  \text{tr}(Z(x) A_d)\right) $$
> > and a convolutional layer output of the form $$H(x) = U^
> > \top Z(x)^\top$$
> > where $A^\star \approx UV^\top$ is a rank-r approximation of $A^\star$.
> > The choice of factorization is simply reflected in Z(x); if I were to choose a different factorization, I would simply take $Z(x) P$ where $P$ is an $m \times m$ rotation matrix. Then, I obtain a new problem
> > $$\min_A \sum L\left(\left( {tr}(Z(x_i)P A_1), \dots,  {tr}(Z(x_i)P A_d)\right) ; y_i\right) \text{ s.t. } \|A\|^*\leq R$$
> > Since the nuclear norm is invariant under orthogonal transformations, $P^\top A^\star$ is also a solution to this problem. Since $PP^\top = I$, this solution yields exactly the same predictor and convolutional layer output. This leads me to conclude that **problems using different factorizations are mathematically equivalent, thus, any difference in performance is due to the inexactness of numerical methods.** But if we run the methods for longer, they should yield the same performance. Am I missing or misunderstanding something here?
> >
> > - Algorithm 2 in the appendix is an ad-hoc method to solve problem (6). It is unclear to me why it is theoretically correct. Intuitively, it makes sense, but there is no proof of convergence. The authors say in their reply: "in solving the dual problem, local minima are global minima," which is clearly correct. However, they never establish that the algorithm converges to a local minimum. In their response, they mentioned other papers that use similar ideas (but for different problems, e.g., https://icml.cc/Conferences/2008/papers/166.pdf). I would like a proof of convergence or a pointer to a proof, similar to Theorem 1 of https://icml.cc/Conferences/2008/papers/166.pdf.

---

> > > ### Author Response · Authors · 2023-12-06
> > >
> > > We thank the reviewer for letting us clarify these points.
> > >
> > > > To make comparisons fair, the authors should explicitly select different ranks for CCNN, display their results, and also report the rank that DCNNN found.
> > > >
> > > > I am still confused about why the choice of factorization makes a difference; I might be missing something.
> > >
> > > We agree with the reviewer in the case that if $m=nP$ but the authors suggest to use $m \ll nP$. In fact, (Zhang et al., 2017) mentions "The computational complexity of each iteration depends on the width $m$ of the matrix $Q$... to improve the computation efficiency, we can use... a tall-and-thin matrix $Q \in \mathbb{R}^{nP \times m}$ such that $K \approx QQ^T$. Typically, the parameter $m$ is chosen to be much smaller than $nP$."
> > >
> > > We understand the confusion and we will clarify this in our manuscript. In our experiments, we used $m=25$.
> > >
> > > Finally, note that in page 2 of our paper, we state "Solving the exact kernel factorization problem itself can be computationally expensive. Looking at Algorithm 1 in the CCNN framework (Zhang et al., 2017), the dimension of the kernel matrix is equal to the number of samples $n$ times the number of convolution operations (patches) $p$. From a space complexity perspective, the factorization step alone occupies memory in the order of $O(n^2 p^2)$. This is hardly feasible: for a small network training on 1000 $28 \times 28$ images with stride 1, i.e. 784 convolution operations, with 64 bit double precision, the kernel matrix takes more than 4 terabytes of memory! Similarly, the computational complexity of many factorization algorithms (for example, Cholesky) can take $O(n^3 p^3)$, which can be catastrophic in practice."
> > >
> > > > Algorithm 2... Intuitively, it makes sense, but there is no proof of convergence.
> > >
> > > By changing maximization to minimization of the negative, and using the extended-value extension (Boyd & Vandenberghe 2004), the dual problem in eq.(6) becomes ${\rm
> > > minimize}\_{\alpha \in \mathbb{R}^n} f(\alpha)$ where:
> > > $$f(\alpha) = \begin{cases}
> > > c \sum_{i=1}^n \ell^*(-\alpha_i/c), & \text{if } \lambda_\max(\sum_{i=1}^n \sum_{j=1}^n \alpha_i \alpha_j y_i y_j K(x_i,x_j)) \leq 1 \text{ and } \alpha_i \geq 0, \forall i \in [n] \\\\
> > > \infty, \text{otherwise}
> > > \end{cases}$$
> > > For instance, for the hinge loss, we have:
> > > $$f(\alpha) = \begin{cases}
> > > -\sum_{i=1}^n \alpha_i, & \text{if } \lambda_\max(\sum_{i=1}^n \sum_{j=1}^n \alpha_i \alpha_j y_i y_j K(x_i,x_j)) \leq 1 \text{ and } 0 \leq \alpha_i \leq c, \forall i \in [n] \\\\
> > > \infty, \text{otherwise}
> > > \end{cases}$$
> > > We can now apply Theorem 4.1 in (Tseng, "Convergence of a block coordinate descent method for nondifferentiable minimization", Journal of Optimization Theory and Applications, 2001. See also Example 6.4 therein.) To be fully formal, we will add an outer loop to our algorithm to go through all coordinates cyclically several times. Currently, we go through all coordinates only once.

---

> > > > ### Comment · Action_Editors · 2023-12-22
> > > > **Further clarifications needed**
> > > >
> > > > Dear authors and reviewers,
> > > >
> > > > I am still trying to fully parse the details of the arguments in this discussion. I have clarity with respect to all points discussed here, except for the point on uniqueness of factorization of the kernel matrix.
> > > >
> > > > As it stands, it is not yet clear to me why the authors' statements are correct: as Rev. BXpE points out, the loss function in Eq. 3 is invariant to rotations matrices (both in loss as in the nuclear norm). As a result, if one employs a rotated factorization for the kernel matrix, then the solution will be a rotated matrix A. Can the authors please clearly explain why this statement is incorrect, o *when* this statement is incorrect?
> > > >
> > > > Thanks,
> > > > A.E.

---

> ### Author Response · Authors · 2023-12-26
>
> Many thanks for allowing us to further clarify this point. Lets consider a simple counterexample for (Zhang et al., 2017) where $n=2$ samples, $P=1$ patches and $m=1$ in Algorithm 1 of (Zhang et al., 2017). Assume the two samples have different labels $y_1 \neq y_2$. Further assume:
>
> $$K = \left\[\begin{matrix}101 & 99 \\\\ 99 & 101\end{matrix}\right\] = \left\[\begin{matrix}10 \\\\ 10\end{matrix}\right\] \left\[\begin{matrix}10 \\\\ 10\end{matrix}\right\]^T + \left\[\begin{matrix}1 \\\\ -1\end{matrix}\right\] \left\[\begin{matrix}1 \\\\ -1\end{matrix}\right\]^T$$
>
> Using $m=1$, assume (Zhang et al., 2017) approximates $K$ with:
>
> $$K \approx \left\[\begin{matrix}100 & 100 \\\\ 100 & 100\end{matrix}\right\] =  \left\[\begin{matrix}10 \\\\ 10\end{matrix}\right\] \left\[\begin{matrix}10 \\\\ 10\end{matrix}\right\]^T$$
>
> Note that $Q = \left\[\begin{matrix}10 \\\\ 10\end{matrix}\right\]$ and $Z(x_i)$ is the $i$-th row of $Q$, and thus $Z(x_1)=Z(x_2)=10$, which would not allow to differentiate between the two samples with different labels $y_1 \neq y_2$. Thus, the approximate factorization of $K$ affects the output of Algorithm 1 of (Zhang et al., 2017), and as mentioned in our previous answer, (Zhang et al., 2017) argues for using $m \ll nP$ due to computational complexity.

---

> > ### Comment · Action_Editors · 2023-12-27
> > **Comments on clarification**
> >
> > Thank you for the example and clarification. Yet, given the previous comment, then my understanding is the following:
> >
> > Both the primal and dual problem are independent of the factorization used for K. However, the computational complexity of the primal is huge in the size of K, which requires a truncation of this matrix and, depending on how this truncation is done, the formulation might depend on the factorization employed. (In passing, why not simply take a best rank-m approximation of K? this would then also be independent of the factorization used). The proposed dual problem here bypasses this issue, as no truncation of K is needed.
> >
> > Is this correct?

---

> > > ### Author Response · Authors · 2024-01-05
> > >
> > > Yes, truncating $K$ as done in the primal problem of (Zhang et al., 2017) creates an issue. The proposed dual problem bypasses this issue, as no truncation of $K$ is needed:
> > >
> > > - As we mentioned in our initial response: "we have experimental evidence for our claim. In our Table 1 in Page 10, we tried 4 different factorizations (of the infinitely many that may exist, as Reviewer BXpE agrees due to rotation) and found that a specific factorization affects accuracy. That is, the specific factorization affects the results of Zhang's CCNN Algorithm 1."
> > >
> > > - In addition (Zhang et al., 2017) suggests to use $m \ll nP$. They mention "The computational complexity of each iteration depends on the width $m$ of the matrix $Q$... to improve the computation efficiency, we can use... a tall-and-thin matrix $Q \in \mathbb{R}^{nP \times m}$ such that $K \approx QQ^T$. Typically, the parameter $m$ is chosen to be much smaller than $nP$."
> > >
> > > - P.S. We believe there was a typo in the AE comment "why not simply take a best rank-m approximation of K? this would then also be independent of the factorization used". It should be "dependent" instead of "independent", as we show in our previous response.
> > >
> > > We also want to add that for kernels that are close to being non-strictly positive definite, solving the primal problem fails to converge. This is somewhat similar in nature to truncation. Next, we present a counterexample for the inverse polynomial kernel, used by (Zhang et al., 2017):
> > >
> > > - The inverse polynomial kernel is $K_{ij} = 1/(2-\gamma x_i^T x_j)$. Consider $\gamma = 0.0001$ and $y_1 = y_2 \neq y_3 = y_4$. Furthermore, let $x_1 = (1/\sqrt{2},1/\sqrt{2})^T$, $x_2 = (-1/\sqrt{2},-1/\sqrt{2})^T$, $x_3 = (1/\sqrt{2},-1/\sqrt{2})^T$ and $x_4 = (-1/\sqrt{2},1/\sqrt{2})^T$. With four-digit precision, we have $K = \left\[\begin{matrix} 0.5000 & 0.5000 & 0.5000 & 0.5000 \\\\ 0.5000 & 0.5000 & 0.5000 & 0.5000 \\\\ 0.5000 & 0.5000 & 0.5000 & 0.5000 \\\\ 0.5000 & 0.5000 & 0.5000 & 0.5000 \end{matrix}\right\]$.
> > >
> > > - The kernel matrix $K$ has eigenvalues $0.000000002, 0.0001, 0.0001, 2.000$. Taking the eigendecomposition $K = U D U^T$, we let $\Phi = U D^{1/2} = \left\[\begin{matrix} 0.0000 & -0.0000 & 0.0050 & 0.7071 \\\\ 0.0000 & 0.0000 & -0.0050 & 0.7071 \\\\ -0.0000 & 0.0050 & 0.0000 & 0.7071 \\\\ -0.0000 & -0.0050 & -0.0000 & 0.7071 \end{matrix}\right\]$ with four-digit precision. Here, the rows of $\Phi$ are $\phi(x_1),\dots,\phi(x_4)$.
> > >
> > > - We verified that the dual problem separates the points perfectly, while the primal problem fails to converge. (We found several other examples with higher values of $\gamma$ but we believe the one presented here is more didactic.)

---

### Review · Reviewer_HJQU · 2023-10-08

**Summary Of Contributions:**

This paper is built upon the existing convexified convolutional neural networks to study the corresponding dual problem from an optimization perspective. The proposal algorithm reduces the computational overhead in constructing the kernel matrix. To overcome the sub-differential of nuclear norms, this paper proposed novel tools for recovering the linear weight and the output.

**Audience:**

Yes

**Claims And Evidence:**

Yes

**Requested Changes:**

See weakness:

1. It would be beneficial to provide theoretical analysis or numerical justification for how the performance varies with different rank values. Additionally, it would be insightful to explain how the improved performance of DCCNNs is derived from finding an optimal rank through numerical experiments or theoretical analysis.

2. We need theoretical explanations or intuitions to understand why DCCNNs outperform CCNNs.

3. Highlighting real-world problems or applications where DCCNNs offer advantages over existing SOTA approaches would help clarify their practical utility and potential impact.

4. Please report the computational time for each algorithm.

5. (Questions.) It's worth noting that solving the nuclear norm problem can often be computationally inefficient. Therefore, it would greatly benefit the paper if the authors could provide more detailed explanations regarding the methods employed in their proposed algorithm.

**Strengths And Weaknesses:**

Strength: The problem formulation is clearly described, and the difference from the existing CCNN is evident.

Weakness:

1. The motivation is not clearly established. It's unclear how performance is affected by the hyperparameter rank.

2. There are no theoretical explanations for why DCCNNs outperform CCNNs.

3. While the performance is shown to be better than CCNN, which has been published for six years, it's challenging to identify its practical applications compared to state-of-the-art network architectures, such as the Transformer.

4. No computational complexity comparison with CCNN.

---

> ### Author Response · Authors · 2023-10-21
>
> > Strength: The problem formulation is clearly described, and the difference from the existing CCNN is evident.
>
> We thank the reviewer for the appreciation of our work.
>
> > Weakness 1. The motivation is not clearly established...
> >
> > Change 1. It would be beneficial to provide theoretical analysis or numerical justification for how the performance varies with different rank values. Additionally, it would be insightful to explain how the improved performance of DCCNNs is derived from finding an optimal rank through numerical experiments or theoretical analysis.
> >
> > Change 2. We need theoretical explanations or intuitions to understand why DCCNNs outperform CCNNs.
>
> Please see "Motivation" in our global response to all reviewers.
>
> > Weakness 3. While the performance is shown to be better than CCNN, which has been published for six years, it's challenging to identify its practical applications compared to state-of-the-art network architectures, such as the Transformer.
> >
> > Change 3. Highlighting real-world problems or applications where DCCNNs offer advantages over existing SOTA approaches would help clarify their practical utility and potential impact.
>
> Note that reviewer 1tmZ appreciates that our DCCNNs are in the ballpark of state-of-the-art results. Moreover, we highlight that TMLR guidelines for reviewers specifically state "it should not be used as a reason to reject work ... because it isn't achieving a new state-of-the-art on some benchmark." We kindly refer to https://jmlr.csail.mit.edu/tmlr/reviewer-guide.html
>
> > Change 4. Please report the computational time for each algorithm.
>
> Note that reviewer BXpE appreciates that our DCCNNs yield computational gains. Please see Appendix G.4 (Discussion on Computational Complexity) in Pages 27-28, which also includes Table 2 in Page 28 (Running time ratio between data and models of different scales) of the Appendix.
>
> > Change 5. (Questions.) It's worth noting that solving the nuclear norm problem can often be computationally inefficient... provide more detailed explanations regarding the methods employed in their proposed algorithm.
>
> Please see "Coordinate Descent" in our global response to all reviewers.

---

### Author Response · Authors · 2023-10-21
**Global response to all reviewers**

# Motivation.

We are glad to have the opportunity to clarify one of the main motivations for our work. We start by clarifying a misconception regarding (Zhang et al., 2017). Please see Algorithm 1 in http://proceedings.mlr.press/v70/zhang17f/zhang17f.pdf

1) Regarding the factorization:

Without getting into the mathematical details first, note that we have experimental evidence for our claim. In our Table 1 in Page 10, we tried 4 different factorizations (of the infinitely many that may exist, as Reviewer BXpE agrees due to rotation) and found that a specific factorization affects accuracy. That is, the specific factorization affects the results of Zhang's CCNN Algorithm 1.

In short, in Zhang's CCNN algorithm: factorization $Q Q^\top$ of K is done in Step 1, such factorization is used (i.e., $Z(x_i)$ are rows of $Q$) is used as input data for finding a low-rank solution in Step 3, eq (12) through the nuclear norm constraint $\\|A\\|\_* \leq R$. That is, the factorization affects the low-rank solution.

Finally, Step 1 of Zhang's CCNN Algorithm 1 does not mention which factorization should be used. In fact, we argue that it is impossible to (optimally) define which factorization, as one would need to have the hindsight that such factorization would lead to a solution of Step 3, eq (12), which is both small in loss (i.e., objective function) and fulfills the nuclear norm constraint.

If needed, we could provide a small illustrative mathematical example (e.g., $n=2$, $p=3$).

2) Regarding the additional parameter (number of filters $r$):

Our DCCNN does not need the additional parameter (number of filters $r$) while Zhang's CCNN does. Note that besides the input data $(x_i,y_i)$ for $i=1$ to $n$ and the kernel $K$ to be used, Zhang's CCNN Algorithm 1 require the regularization parameter $R$ and the number of filters $r$. Our Algorithm 2 in Appendix E.1 requires the regularization parameter $c$. Note that $R$ and $c$ are similar in nature (constraint $\\|A\\|\_* \leq R$ in Zhang's CCNN, penalty $c \\|A\\|\_*$ in our DCCNN). Thus, one can easily see that there is an additional parameter (number of filters $r$) in Zhang's CCNN, which Zhang's CCNN Algorithm 1 uses in Step 4 "Compute a rank-r approximation" of $\hat{A}$.

In contrast, in our DCCNN, the rank is automatically computed by considering the eigenvalues $\lambda_i=1$ (Theorems 2 and 5).

# Coordinate Descent.

We perform coordinate descent optimization of the dual problem in eq.(6). Please see Algorithm 2 in Appendix E.1.

First, note that the primal and dual problem are convex optimization problems. Thus, in solving the dual problem, local minima are global minima. Any reasonable method will converge to the optimum (e.g., gradient descent, coordinate descent, interior point, etc.)

We initially tried an interior point method for solving the dual problem, but it was too slow and required a lot of memory. We decided not to report this. We then chose a coordinate descent method for solving the dual problem. This idea has been used before in prior literature in other problems, see e.g.,

[1] Hsieh et. al. A Dual Coordinate Descent Method for Large-scale Linear SVM. ICML 2008.

[2] Chang et. al. Dual Coordinate Descent Algorithms for Efficient Large Margin Structured Prediction. TACL 2013.

We tried the regular coordinate ordering ($i = 1$ to $n$) and a random permutation of coordinates, but we found that sorting according to $\lambda_\max(K(x_i,x_i))$ worked better experimentally as well as made some intuitive sense from the constraint of the problem. Having said that, any other ordering would have worked, albeit with a slower convergence.

---

### Comment · Action_Editors · 2023-10-22
**Discussion period active**

Dear Reviewers,

Please take some time to review the authors' response to your initial questions and comments, and seek to clarify any outstanding doubts.

A.E.

---

### Decision · Action_Editor_DwbX · 2024-01-08

**Recommendation:** Accept with minor revision

**Comment:**

This paper proposes an interesting alternative to the method of (Zhang et al., 2017) by studying the dual problem of convexified convolutional neural networks. Most reviewers agree that this is a valuable contribution to the field, and are satisfied with the general manuscript and empirical results. One reviewer had special concerns regarding the comment, made throughout the manuscript, that the solution to the primal problem depends on chosen factorization for the kernel matrix. This has led to insightful discussions among reviewers and authors.

Based on this discussion, I am recommending this paper be accepted with the following necessary revisions:
* The comments and suggestions made by all three reviewers should be incorporated, as the authors mentioned they would.
* Specific attention should be given to the motivation for this work based on the discussions arising from Reviewer BXpE. More precisely: the primal problem does not depend on the factorization of the kernel matrix (as the loss is invariant to it). The dependence on the factorization only appears once this factorization is truncated, which is needed for computational constraints (and thus the *algorithm* of Zhang et al does depend on the factorization chosen). The dual problem proposed in this work bypasses this limitation, thus representing a valuable alternative.
* The manuscript should be revised for typos and format (the authors are highly encouraged to include a Concluding section that summarizes their contributions given prior work).

**Audience:**

The contribution in this paper will be of relevance to the TMLR community at large.

**Claims And Evidence:**

The claims made in this paper have been made clear and supported by evidence during the discussion period. These should be made clear in the revised version of this paper.